# Safe and Effective Treatment of Compromised Clavicle Fracture of the Medial and Lateral Third Using Focused Shockwaves

**DOI:** 10.3390/jcm11071988

**Published:** 2022-04-02

**Authors:** Rainer Mittermayr, Nicolas Haffner, Sebastian Eder, Jonas Flatscher, Wolfgang Schaden, Paul Slezak, Cyrill Slezak

**Affiliations:** 1Department Meidling, AUVA Trauma Center Vienna, 1120 Vienna, Austria; sebastian.eder@auva.at (S.E.); med.eswt.schaden@aon.at (W.S.); 2Ludwig Boltzmann Institute for Traumatology, 1200 Vienna, Austria; n.haffner@osteodoc.at (N.H.); contact.jf@posteo.de (J.F.); paul.slezak@trauma.lbg.ac.at (P.S.); cyrill.slezak@gmail.com (C.S.); 3Austrian Cluster for Tissue Regeneration, 1090 Vienna, Austria; 4Department of Physics, Utah Valley University, Orem, UT 84058, USA

**Keywords:** shockwave therapy, regenerative medicine, clavicula, compromised fracture healing

## Abstract

A delay or failure to heal is the most common possible complication in clavicle fractures, especially in cases primarily treated conservatively. As the current standard therapy, surgical revision achieves good healing results, but is associated with potential surgery-related complications. Shockwave therapy as a non-invasive therapy shows similar reasonable consolidation rates in the non-union of different localizations, but avoids complications. Compromised clavicle fractures in the middle and lateral third treated with focused high-energy shockwave therapy were compared with those treated with surgical revision (ORIF). In addition, a three-dimensional computer simulation for evaluating the pressure distribution during shockwave application accompanied the clinical study. A comparable healing rate in bony consolidation was achieved in both groups. Significantly fewer complications, however, occurred in the shockwave group. The simulations showed safe application in this instance, particularly in avoiding lung tissue affection. When applied correctly, shockwaves represent a safe and promising therapy option for compromised clavicle fractures in the middle and lateral third.

## 1. Introduction

Fractures of the clavicle are common (5 to 10% of all fractures [1]) and have shown an increasing incidence, especially over recent years. Mainly active young men are at risk of suffering a fracture in the diaphysis (type I, according to the classification of Allmann, which accounts for approximately 80% of all clavicle fractures [2,3,4]) in sports accidents, or with bicycles or motorcycles [5].

Recently, studies have shown inferior outcomes and increased complications, particularly non-unions, after conservative treatment compared to surgical intervention [6,7,8,9]. This contrasts with the former propagation of better results by applying conservative treatment along with sufficient analgesic therapy and transient immobilization (e.g., figure-of-8 bandages, sling, cuff and collar) [10,11]. However, especially in the first rehabilitative phase, surgical treatment shows advantages over conservative therapy due to the earlier possibility of mobilization [9,12,13]. Considering the high number of cases in active young male patients, the time that elapsed before patients could return to sports was significantly longer in patients treated conservatively than in patients who underwent surgery [9,13,14,15]. In addition to the standard indications for surgery of acute clavicular fracture (i.e., open fracture, complicated soft tissue condition, neurovascular complication), length shortening exceeding 10%, age, activity level, and dominant hand are considerations that have become increasingly discussed as a relative indication for surgery [16].

The most common surgical methods include intramedullary fixation and plate osteosynthesis [17]. Equivalent functional outcomes and union rates can be achieved with indirect reduction and titanium elastic-nail fixation compared to open reduction and plating. Reduced surgery time [18], small incisions with more minor disturbances of sensitivity, and minimal tissue damage are potential advantages of the indirect technique [17,19,20,21]. On the other hand, aspects such as telescoping, secondary shortening, and pin migration/perforation must be considered [22]. Possible risks associated with open reduction and internal fixation (ORIF) are neuropathy in the vicinity of the incision area, infection, implant failure, pneumothorax, the necessity for hardware removal if irritation occurs, and weakened clavicular biomechanics after removal of the osteosynthetic material [23,24,25,26,27].

However, the most critical complication following clavicle fractures is the development of fracture non-union, which is reported in up to 24% of those treated conservatively [9,28]. In contrast, there are consistent reports of much lower non-union rates of approximately 3% in patients who received surgical treatment [29,30]. In addition, general risk factors, independent of primary care, that may result in non-union include fracture-inherent factors such as an open fracture; comminuted fracture; shortening by more than 2 cm or 10%; and patient-related factors such as age, gender, and smoking [24,29].

Not all non-unions are necessarily symptomatic (tight pseudarthrosis). However, if this is the case, pain and functional limitations of the shoulder girdle are the main problems reported. Consequently, symptomatic pseudarthrosis is primarily surgically treated. Convincing healing results can be achieved by applying different plate configurations and plate localizations, usually combined with bone grafts [31,32,33,34]. However, a decisive disadvantage is the invasive nature of the intervention in clavicle non-union revision surgery, which is associated with all common complications of surgery. These may include, but are not limited to, plate loosening or implant failure, infection, sensory disturbance, shoulder stiffness, the need for removal if the osteosynthesis material is symptomatic, and the risk of refracture after removal [31,35,36].

This study proposes that focused high-energy extracorporeal shockwave therapy (ESWT), which has emerged in treating other fracture non-unions or delayed unions, is a non-invasive and highly effective treatment alternative to surgery [37,38,39]. The application of the focused extracorporeal shockwave induces molecular and cellular processes through mechanotransduction, which supports tissue regeneration without causing lesions [38,40,41]. The treatment of delayed or fracture non-unions employing focused, high-energy shockwaves can be realized in different anatomic areas, with a favorable complication–benefit ratio. Moreover, treatment-associated costs can be reduced by considering ESWT [42,43].

Regarding the safe application of ESWT to the clavicle in the vicinity of lung tissue, a series of considerations has to be made. The most recent Consensus Statement on ESWT Indications and Contraindications [44] lists lung tissue in the area of high-energy focused shockwaves as a contraindication. This avoidance of lung tissue in the focal area is based on the destructive properties of tensile waves, the primary cause of observed pulmonary capillary hemorrhage. Peak negative pressures tend to be relatively small in ESWT, at about 10–20% of peak positive pressure [45]; however, to safely apply shockwaves close to the lungs, it is essential to understand their impact on the tissue.

While the underlying mechanisms inducing pulmonary capillary hemorrhage have been systematically studied for ultrasound [46], comparably few documented concerns relating to shockwave treatments are reported [47]. The present-day safety requirements of diagnostic ultrasound imaging at or in the vicinity of the lung are built around the thermal index (TI) and the mechanical index (MI), which remains applicable for shockwaves. Both parameters consider worst-case estimates to limit the risk of bioeffects associated with inertial cavitation effects [48]. TI is defined as the ratio of the in situ acoustical power and the power necessary to raise the tissue temperature [49]. MI is defined as the ratio of the peak in situ refractional pressure and the square root of the center frequency. However, utilizing the TI and MI, including non-thermal mechanisms, results in an incomplete picture of pulmonary safety, ignoring potentially important acoustic parameters [50].

Clearly defined safety parameters of an MI ≤ 1.9 exist for human applications by the Food and Drug Administration. In addition, multitudes of studies have provided experimental data on pre-clinical animal trials [51]. However, to adapt these findings to the potential use of ESWT, we find that many of the discussed parameters do not apply. For one, the TI described localized heating associated with shockwave pulses with an associated application frequency of <10 Hz as insignificant compared to continuous ultrasound imaging modalities at comparable peak pressures. Secondly, any MI-threshold modeling parameters, such as center frequency, pulse repetition frequency, pulse duration, and exposure duration, cannot be translated to singular shockwaves. This is especially true for any frequency dependency, as typical shockwaves span a spectral profile ranging from a few Hz to low MHz [45]. This may, however, be less problematic, as Miller et al. have shown frequency independence of peak tensile-pressure thresholds for pulmonary capillary hemorrhaging in rats in the range of imaging frequencies [52]. An additional minor dependence on a spatial-peak pulse-average was observed, but these also involve ultrasound-specific temporal averages of all propagating pulses.

Peak tensile-pressure thresholds for ultrasonic studies relating to pulmonary safety rely on approximating in situ acoustic fields as direct measurements, which are not possible within or close to the lungs due to absent effective couplings. These are commonly obtained by utilizing a 0.3 dB [(cm·MHz)]^−1^ free-field attenuated water-bath reference measurement of the transducer. For focused ESW applicators, these attenuated values may vary significantly due to the dephasing of the acoustic waves in tissue inhomogeneities, but provide a solid upper threshold estimate. Ultimately, only clinical studies will provide definitive safety parameters. However, the combination of extensive pre-clinical animal testing and established reviews finding no consistent risk for diagnostic lung ultrasounds [53] provide confidence in applying peak tensile-pressure thresholds to obtain ESWT safety in near-lung applications.

## 2. Materials and Methods

### 2.1. Clinical Study

At our first-level trauma center, a prospective open clinical study is being conducted on the efficacy of shockwave therapy in compromised, delayed, or non-union fractures in various locations. Based on this data, we performed a retrospective monocentric study comparing ESWT-treated clavicular fractures with surgery, the current commonly accepted standard of care. We considered cases with or without autologous bone graft followed by plating, from 1999 until 2018. The IRB approved the study (AUVA 12/2018). Consequently, the institutional surgery database was screened for revision surgeries of clavicle fractures (regardless of initial primary care). These surgical cases were then compared with those treated, non-invasively, using focused electrohydraulic high-energy shockwaves.

A failure in fracture-healing was defined as missing bony consolidation of at least 2 of 4 cortices in standard anterior–posterior and lateral X-ray views. Non-union was defined as the failure in fracture healing within six months despite adequate primary surgical and conservative interventions. A delayed union, correspondingly lacked bridging within 3 to 6 months after trauma.

#### 2.1.1. Shockwave Group

Patients (*n* = 28) were under general anesthesia provided by a larynx mask, intravenous sedation and analgesia. They were in a supine position, slightly elevated, and their heads turned in the opposite direction to the affected clavicle. According to Allmann I and II (Neer classification Type I), clavicle fractures in the diaphysis and lateral aspect were included, regardless of whether they were initially treated conservatively or surgically.

X-ray fluoroscopy was used first to identify the location of failed union, which was marked on the skin using a permanent marker. Afterward, an X-ray was positioned in a manner that stimulated the target focus of the shockwave therapy head, in order to orient it in the right direction and avoid lung affection (see Figure 1).

On the marked area, abundant bubble-free conduction medium was applied. The therapy head was positioned according to the previously determined angle using X-ray, focusing on the fracture site (Figure 1). All shockwave treatments were performed with an electrohydraulic device (Orthogold 280C, MTS medical UG, Konstanz, Germany). In total, 3000 impulses were applied, with an energy flux density of 0.4 mJ/mm^2^ (−6 dB) and a frequency of 4 Hertz. The direction of application was changed after half of the pulses.

During the application, the anesthesiologist continuously monitored ventilation parameters—in particular, exhaled carbon dioxide, oxygen saturation, and ventilation resistance—to rule out lung affection. Post-interventional lung X-rays should exclude pneumothoraces.

A Gilchrist sling subsequently immobilized all clavicles for 3 to 4 weeks. After that, until two months after treatment, physiotherapy without load was allowed up to 60 degrees. The patients then performed continued free exercise without weight-bearing for up to 3 months.

#### 2.1.2. Surgery Group

Patients (*n* = 21) suffering from symptomatic clavicular unhealed fractures received ORIF revision surgery. First, resection of the interfragmentary scar and fracture fragment ends was performed until clinical vital bone tissue was evident. In patients who initially underwent surgery for their acute clavicle fracture, metal removal was performed first during the surgical revision procedure. Afterward, recanalization of the medullar cavity was performed using a 2 mm drill. Subsequently, open reduction and plating were performed. In most cases (*n* = 17), an additional autologous bone graft from the iliac crest was performed.

All but one patient received a Gilchrist sling for 2 to 6 weeks for immobilization. The same rehabilitation program was established as in the shockwave group.

#### 2.1.3. Outcome Parameter

The study’s principal endpoint was bony healing at 3 and 6 months. An independent blinded clinician evaluated radiographs at these time points for healing progression. Bony consolidation was diagnosed if at least 3 of 4 cortices on conventional radiographs showed bony bridging and patients were free of symptoms. Computed tomography was performed if uncertainties remained in the radiographs.

Demographic data were recorded, and the potential influence on the healing was assessed. Moreover, any complications associated with the study therapy were documented and evaluated.

### 2.2. Data Analysis

All data were collected and organized using Microsoft Office Excel software Version 2021. Quantitative data are shown with uncertainties ± standard deviations and range (min, max). Pair-wise comparisons were based on a two-tailed, heteroscedastic Student’s *t*-test. Categorical factors and their associations were studied using a two-sided Fisher’s exact test using the method of summing small *p* values (GraphPad Prism version 9.0.2 for Windows, GraphPad Software, San Diego, CA, USA). The results were considered statistically significant when the *p*-value was lower than 0.05.

### 2.3. Computer Simulation

To assess safety and evaluate the ESWT pressure field during therapy, a comprehensive three-dimensional computational simulation was performed. There is no experimental approach available that allows for the in situ mapping of sound fields during ESWT application. In lieu of this, a computer simulation can provide insights into wave propagation and pressure zones. Shockwave pressure fields within the therapy zone were calculated using the MATLAB toolbox k-wave, using a spatial grid spacing of 0.5 mm and a time step-size of 27 ns. This yielded a maximum supported frequency of 1.48 MHz. The total simulated volume was 246 mm × 152 mm × 152 mm, in addition to a PML with a thickness of 10 grid points on each border. Simulations were run on an NVIDIA^®^ GTX 1070 graphics card using the CUDA implementation of k-Wave, recording the maximum and minimum pressure over the whole domain, and the pressure signal at critical positions close to the clavicle and the lung.

Acoustic tissue parameters were derived from CT image data from the shoulder of the Visible Human Project^®^, with a resolution of 0.5 mm. Hounsfield attenuation values were converted into material-density and speed-of-sound values using the k-Wave function hounsfield2density based on Schneider et al. [54]. For the attenuation values, the tissue was differentiated into muscle ($\alpha_{0}$ = 0.93 $\mathrm{dB}/\left({\mathrm{MHz}}^{y}\mathrm{cm}\right)\;$, B/A = 7.5); fat ($\alpha_{0}$ = 0.51 $\mathrm{dB}/\left({\mathrm{MHz}}^{y}\mathrm{cm}\right)\;$, B/A = 10); bone ($\alpha_{0}$ = 11.91 $\mathrm{dB}/\left({\mathrm{MHz}}^{y}\mathrm{cm}\right)\;$, B/A = 0); and water ($\alpha_{0}$ = 2 × 10^−3^ $\mathrm{dB}/\left({\mathrm{MHz}}^{y}\mathrm{cm}\right)\;$, B/A = 5.2), using a power law factor of y = 1.2.

The simulated elliptical reflector was based on the geometry of the actual electrohydraulic MTS orthogold280C used in the clinical study, using an input-source pressure-wave profile yielding a computed peak-focus reference pressure of 30 MPa in the free-field.

## 3. Results

In the present study, 49 fractures of the clavicle showing disturbances in healing were included. Most of them occurred in the medial third (*n* = 35) and only 14 in the lateral aspect.

The ratio between the shock-wave group and the surgical group was 28 to 21 patients, with a similar gender distribution (see Table 1). Considerably more male patients were injured in a traffic accident or during sports activities (e.g., cycling, skiing), but there was no difference between the two groups. The patient population in the present study was also rather young, as was expected and in line with the reported literature. A substantially higher number of lateral fractures was located in the shockwave treatment group than in the operative group (*p* = 0.0126), in which most of the treated cases affected the middle third.

While the shockwave group showed a homogeneous distribution between hypertrophic and atrophic cases, the distribution in the operative group was two-thirds in favor of the hypertrophic issues.

Most patients were referred to our clinic to treat an unhealed clavicle fracture (*n* = 33). In this context, early treatment was performed in three (51 to 64 days after accident) and two (66 and 80 days after initial trauma) cases in the shockwave group and operative group, respectively, without complying with the time criteria for a delayed union.

Between 3 and 6 months (meeting the definition of delayed healing), 11 (39%) and 5 (24%) cases were in the shockwave therapy group and surgical group, respectively. The remaining cases were all older than six months and involved 15 fractures in the shockwave group (54%) and 14 cases in the operative group (67%). No relevant co-morbidities were noticed in either study group.

All acute conservative fractures were treated with a figure-of-eight bandage. All primary operative cases received an open reduction and internal fixation with a plate. In the previous treatments, surgical repair was performed in all cases (*n* = 4 in the ESWT group and *n* = 4 in the operative group).

Healing at three months (see Table 2), implying bridging of at least three of four cortices on standard radiography, was noted in only 46% of the shockwave group. Similarly, clavicle fractures after revision surgery showed bony consolidation in only 43%, which is not statistically significantly different (*p* > 0.9999).

Conversely, at six months, 75% of the shockwave-treated patients showed healing of their clavicle fracture. Comparable results (*p* = 0.7172) were obtained with surgery, with a 71% bony consolidation rate (see Figure 2). In the ESWT group, all patients could be followed up at both 3 and 6 months. This was possible in the surgical group only after three months. Unfortunately, we lost three patients to follow-up in the 6-month interval. However, it is essential to mention that two patients already showed healing after only three months. If one also declares these two patients as cured after six months, one arrives at an overall cure rate of 81%.

Notably, eight fractures that did not show healing at the 3-month time point were healed after six months after a single shockwave treatment. Equally, for the surgical intervention, six months did pass in eight of the cases before complete healing could be diagnosed.

Before reaching the definition of delayed healing, three months, the early intervention completely restored the cortical continuity of the fracture in both groups. In the fractures that showed delayed healing (between 3 and 6 months), those in the operative group showed 100% complete healing (5 out of 5). In the shockwave group, 73% (8 of 11) showed healing in this entity. Those three fractures that did not heal showed a pronounced diastasis of more than 5 cm.

Considering the operation and treatment times, ESWT was completed in an average of 18 ± 13 (11–83) minutes, with 3000 pulses at a frequency of 4 Hertz. On the other hand, the surgical intervention (bone grafting, plating) required significantly (*p* < 0.0001) more time, at 124 ± 55 (80–282) minutes.

Looking at the complication rates for the different forms of therapy, ESWT was not associated with a single complication in all cases. In particular, no signs of lung-tissue injury or postinterventional respiratory problems were evident in any patients receiving shockwaves to either the medial or lateral third of the clavicle. In contrast, the surgical group had four serious complications (19%). One patient had a revision-worthy hematoma at the cancellous-bone-harvest site. Furthermore, one plexus lesion, one irritating implant, and sensory disturbances in the pelvic region were recorded.

Figure 3 depicts a simulated representative applied pressure field about the treatment zone focused on the clavicle. Attenuated peak pressures reach a maximum value of pmax = 23.95 MPa at the bone. The focal volume is slightly deformed, and an apparent deflection is seen alongside the bone in the cranial direction. There is no significant transmission through the bone.

Figure 4A–C takes a closer look at the tensile pressure distribution. We see peak negative pressures in the close vicinity of the clavicle, which rapidly falls off with distance. A similar deflection to the peak positive pressures about the bone interface can be observed for the negative pressures [55]. Taking a closer look at the waveform recorded at the surface of the lung tissue, corresponding to the most considerable observed tensile pressure pmin = −0.68 MPa (Figure 4E), we see the expected enhanced negative pressure due to the reflection at the tissue/air interface. Notably, both the positive and negative pulse widths are of comparable duration in the far field over the focal reference (Figure 4D).

## 4. Discussion

In this study, we were able to show that the application of high-energy shockwaves for the treatment of complicated clavicle fractures can be performed safely, and that outcomes were comparable to surgical intervention with an absence of complications commonly seen after surgery. Pressure simulations show that the lungs are not endangered during the treatment if performed correctly (correct direction of application). We were also able to demonstrate this in the present clinical study. No single complication during the application was encountered. Furthermore, we achieved a 75% cure rate comparable to that of surgical intervention.

Clavicle fractures are among the most common injuries, with a bimodal distribution: young males from sports injuries or the elderly from a trivial fall. Incomplete healing and the formation of a non-union are some of the most relevant complications related to this entity, especially in the conservative primary care of displaced fractures [6,9,28,56,57]. Even if surgery shows better results in terms of healing, a critical evaluation should be made based on patient claims, and conservative treatment should not be ruled out per se. Even if an initial surgical intervention achieves better results than conservative treatment, potentially severe complications need to be considered. In fact, one study shows that revision surgery for failure to heal with primary conservative treatment is not a predictor of increased complications [58].

In approximately two-thirds of patients with clavicle non-union, surgery is performed due to persistent symptoms [59]. In addition, surgical treatment of clavicular fracture non-union has achieved excellent healing results, depending on the reported literature [31,32,33,36]. Our study achieved a healing rate of 71% in the surgical intervention group. Unfortunately, three patients (14%) were lost to follow-up, of which two had already healed after three months. Moreover, our patient population was demanding, as evidenced by the number of surgical interventions due to complicated healing prior to study-specific surgery. Three patients (14%) were operated on at least once because of the failure to heal beforehand.

The same is true for the group that received shockwave treatment. Although all patients could be followed over the entire follow-up period, four patients (14%) were surgically revised at least once for failure to consolidate before the study-specific treatment with shockwaves was scheduled.

However, even in light of promising healing results, one must not forget the severe surgery revision complications that may sometimes arise, both at the clavicle and the site of harvest of the cancellous bone. Literature reports about common complications include, but are not limited to, failure of the osteosynthesis, symptomatic hardware with the necessity of removal, infections, hematoma, disturbances of sensitivity, and lesions of the brachial plexus [35,60,61,62,63,64,65]. Furthermore, despite the excellent results achieved, surgical revision cannot guarantee healing in compromised fracture healing of the clavicle [63,66].

Shockwave treatment, however, represents a safe, effective, and cost-saving method, which is increasingly considered a first-line treatment in fracture non-unions [38,39]. Currently, the International Society for Medical Shockwave Treatment (ISMST, www.shockwavetherapy.org, accessed on 9 March 2022) advises against treatment over lung tissue because energy absorption occurs at the interface of tissues of different densities, and potential tissue damage could occur. However, we show in this study that the application can, indeed, be performed safely on the medial and lateral aspects of clavicles.

The simulation results also clearly show that a tightly focused peak pressure remains compact in the therapy zone. After that, there is a rapid drop-off in peak pressure with distance from the focal point until pressure variations reach the lungs. At this point, peak pressures are small, but the reflection about the interface of air with the lungs does enhance the tensile wave.

In ascertaining from the simulation whether there is potential for lung-tissue damage, we only look at the single voxel of maximum peak tensile pressure to obtain an upper bound. In determining the MI associated with the incident wave, it is necessary to determine a center-frequency equivalent. We derive an effective frequency for the shockwave from the negative half-cycle period of the pressure wave [67]. This yields a frequency of f = 139.9 kHz for the pressure wave shown in Figure 4E. Ultrasound thresholds are mostly applied for devices with frequencies of 0.5 MHz and above. Ahmadi et al. [68] propose a modified mechanical model which addresses the mechanical index’s divergence for low frequencies (LF). In the proposed form ${\mathrm{MI}}_{\mathrm{LF}}\frac{\left(p\;-{\;p}_{0}\right)}{\sqrt{f}}$, where the ambient pressure *p_0_* is included, we obtain a MI_LF_ of 1.54. In comparison, the traditional MI = 1.81 still yields a value below the FDA threshold. It is worth mentioning that this is a worst-case scenario at the singular voxel, and is considered for two main reasons. For one, the high-frequency cut-off due to computational limitations results in more energy being carried in the lower frequencies. The missing high-frequency components would be attenuated disproportionately faster in the biological system, leading to a further reduction in in situ pressures. Secondly, this limitation also impacts the input signal, which is longer than the experimental pulse width. This results in a lower-than-expected center-frequency equivalent, which leads to a higher mechanical index. These overly conservative approximations lead us to conclude that no adverse effects on lung tissue associated with an MI threshold can be expected in the simulated application.

Clinically, continuous and close anesthesiologic monitoring did not reveal any respiratory abnormalities, and shockwaves induced no pneumothorax. However, one must be aware, when offering patients the option of ESWT, that no correction of an existing bony malposition can be achieved. This, in turn, requires that the course of therapy be discussed with the patient [69]. If the (young) patient has high physical demands (i.e., they are an athlete), and the resulting scapulothoracic rhythm would be substantially impaired, surgical revision should be considered. On the other hand, shockwave treatment can prevent complications associated with surgery while showing comparable healing outcomes [37,39,70]. Moreover, aftercare is not different from that undergone by operated patients, but the duration of hospitalization is substantially reduced. Patients treated with ESWT can leave the hospital the day after treatment, which is rarely the case with surgery. Thus, the inpatient costs can be drastically reduced, as can the operation time and material expenditure.

This study includes limitations due to the retrospective nature of the analyses. Some data within the parameters are not homogeneously distributed between the study groups (fracture location, fracture etiology), leading to limited comparability. However, comparing only medial fractures, both groups showed similar results (69% vs. 74% in shockwave therapy vs. the operative group). Considering the healing response in the different etiologies (atrophic vs. hypertrophic), no differences were observed within nor between the study groups. Another limitation is the small number of cases within the study, although a long period was taken into account. We also included patients with impaired fracture healing before the temporal definition of delayed healing (shockwave, *n* = 3; surgery, *n* = 2). All fractures healed in both groups. Methodologically, it is not possible to measure the pressure distribution of ESWT directly at the application site or in the lungs. However, our computational simulation indicated that no harmful pressures were applied to the lungs, which is in agreement with and underlines what was observed clinically in all patients in terms of the absence of complications.

## 5. Conclusions

This study, which is comprised of a simulation and a clinical evaluation, shows that shockwave treatment (ESWT) is a safe application for mid- and lateral-clavicle fractures. ESWT shows good healing results comparable to the surgical treatment of compromised clavicle fractures, while avoiding surgery-related risks and complications. Additionally, the use of shockwaves in these scenarios may help to save health-care expenditure by reducing the costs of materials and hospitalization.

## Figures and Tables

**Figure 1 jcm-11-01988-f001:**
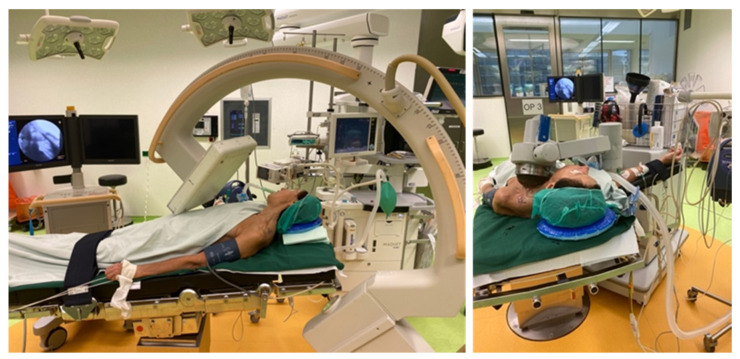
Pre-operative evaluation of the fracture site and determination of the direction of application (**left**). The shockwave therapy head is positioned on the clavicle region (**right**).

**Figure 2 jcm-11-01988-f002:**
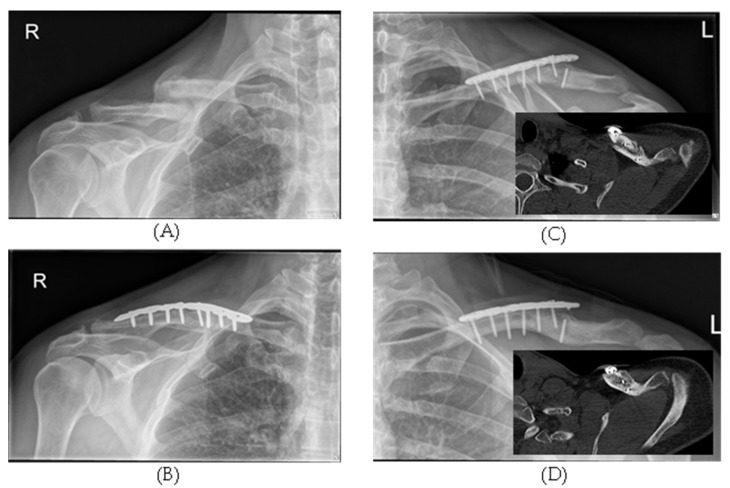
(**A**) Non-union of the right clavicle in a 51-year-old male. (**B**) Six months post-surgery, bony healing was achieved using standard-of-care treatment with non-union interfragmentary scar resection, autologous bone graft, and plating; (**C**) a 26-year-old male patient who received plating of a mid-third clavicle acute fracture showed a non-union associated with a hardware failure. The CT scan (insert) confirms the indirect signs of non-union (i.e., screw failure, implant loosening). The patient refused revision surgery and focused high-energy shockwaves were applied. Six months after treatment, bridging occurred (**D**), confirmed by another CT scan (insert).

**Figure 3 jcm-11-01988-f003:**
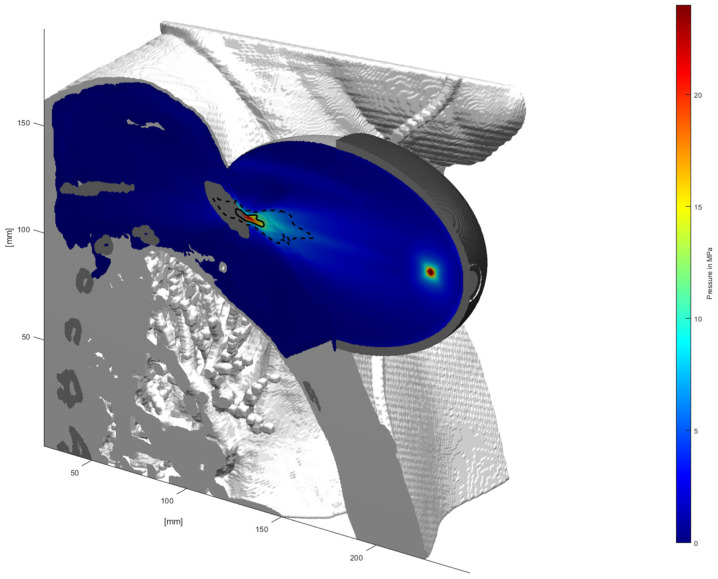
Peak positive pressure distribution: −6 dB and 5 MPa focal zones are delineated by solid and dashed lines.

**Figure 4 jcm-11-01988-f004:**
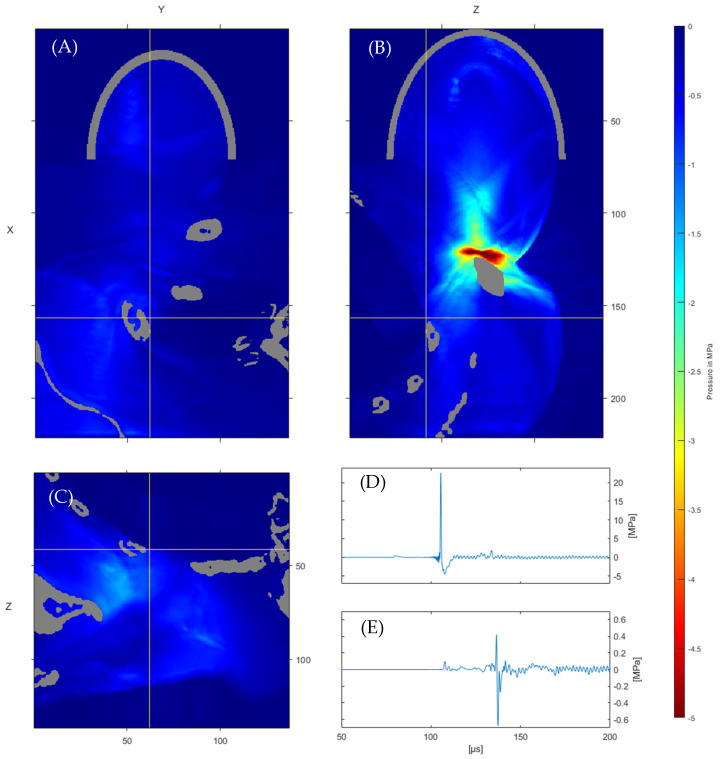
Peak tensile pressure distribution (**A**–**C**) with spatial units (mm) and waveform at the reference focal point (**D**), and the indicated position of maximum tensile stress at lung surface (**E**).

**Table 1 jcm-11-01988-t001:** Patients’ demographics and fracture details. The *p*-value indicates statistical significance between the ESWT and Surgical groups.

Parameter	ESWT (*n* = 28)	Surgery (*n* = 21)	*p*-Value
**Age**	44 ± 13 (15–75)	43 ± 12 (21–62)	0.750
**Gender**			
Female	8 (29%)	8 (38%)	0.5474
Male	20 (71%)	13 (62%)	
**Initial treatment**			
Conservative	18 (64%)	16 (76%)	0.5327
Surgical	10 (36%)	5 (24%)	
**Location**			
Medial	16 (57%)	19 (90%)	0.0126 *
Lateral	12 (43%)	2 (10%)	
**Etiology**			
Atrophic	15 (54%)	7 (33%)	0.2460
Hypertrophic	13 (46%)	14 (67%)	
**Time—trauma to intervention**	279 ± 205 (51–905)	232 ± 108 (66–541)	0.3170
**Previous treatments**	4 (1–2)	2 (1–2)	

* indicates statistical significant more lateral fractures in the ESWT group compared to the surgical group.

**Table 2 jcm-11-01988-t002:** Patient outcomes over time and statistical significance between the ESWT and Surgical groups are indicated by the *p*-value.

Outcome	ESWT (*n* = 28)	Surgery (*n* = 21)	*p*-Value
**After 3 months**			
Healed	13 (46%)	9 (43%)	>0.9999
Not healed	15 (64%)	12 (57%)	
**After 6 months**			
Healed	21 (75%)	15 (71%)	0.7172
Not healed	7 (25%)	3 (14%)	
Lost follow up	-	3 (14%)	
**Complications**			
Complications	0 (0%)	4 (19%)	0.0282 *
No complications	28 (100%)	21 (81%)	

* indicates statistical significant more complications in the surgical group compared to the ESWT group.

## Data Availability

The data presented in this study are available on request from the corresponding author. The data are not publicly available due to ethical and privacy restrictions.

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
