# Peer review of "Safe and Effective Treatment of Compromised Clavicle Fracture of the Medial and Lateral Third Using Focused Shockwaves"

_jcm, 2022, doi:10.3390/jcm11071988_

Round 1
Reviewer 1 Report
This paper provides insight into alternative to ORIF treatment of clavicle fractures. It is well prepared and executed study. The only concern is that there was a significant difference in number of atrophic vs hypertrophic non-unions in the surgery group comparing to ESWT. This should be mentioned in discussion whether it had any impact on the results.
Author Response
We are grateful to the reviewers comment. We added the following sentence (highlighted in yellow) to the limitation section:
This study includes limitations due to the retrospective nature of the analyses. Some data within the parameters is not homogeneously distributed between the study groups (fracture location, fracture etiology), leading to limited comparability. However, comparing only medial fractures, both groups showed similar results (69% vs. 74% in shock wave therapy vs. operative group). Considering the healing response in the different etiologies (atrophic vs hypertrophic), no differences were observed within but also between the study groups.
Reviewer 2 Report
Shockwave therapy is so far not well described as aid in healing of clavicular fractures.
Available literature suggests potential benefit, but is mainly based on case reports and one prospective study.
Since this is prospective study is definitely of high importance for the field. Follow-up time is appropriate for expected healing.
Presented tables and images support described results. Findings are important for everyday practice, due to the possibility to
avoid re-operation by using this technique.
In materials&methods please write exact number of patients (n) in "Schockwave group". Otherwise well structued article with introduction describing many details. This article will contribute significantly to orthopaedic surgeons.
Author Response
We want to thank for the reviewers kind comments!
We added the number of patients accordingly.